# The Tripartite Lichen *Ricasolia virens*: Involvement of Cyanobacteria and Bacteria in Its Morphogenesis

**DOI:** 10.3390/microorganisms11061517

**Published:** 2023-06-07

**Authors:** Francisco J. García-Breijo, Arantzazu Molins, José Reig-Armiñana, Eva Barreno

**Affiliations:** 1Departamento de Ecosistemas Agroforestales, ETSIAMN, Universitat Politècnica de València, Camino de Vera s/n, 46022 València, Spain; fjgarci@upv.es; 2Instituto Cavanilles de Biodiversidad y Biología Evolutiva (ICBiBE), Botánica, Universitat de València, C/Dr. Moliner, 50, 46100 Burjassot, Spain; arantxa.molins@gmail.com (A.M.); jose.reig@uv.es (J.R.-A.); 3Instituto de Investigaciones Agroambientales y de Economía del Agua (INAGEA), Departamento de Biología, Universitat de les Illes Balears (UIB), Ctra. Valldemossa Km.7., 07122 Palma de Malllorca, Spain

**Keywords:** cephalodia, endangered lichens, non-photosynthetical bacteria, *Nostoc*, *Symbiochloris*, TEM, tripartite lichen

## Abstract

*Ricasolia virens* is an epiphytic lichen-forming fungus mainly distributed in Western Europe and Macaronesia in well-structured forests with ecological continuity that lack eutrophication. It is considered to be threatened or extinct in many territories in Europe (IUCN). Despite its biological and ecological relevance, studies on this taxon are scarce. The thalli are tripartite, and the mycobiont has a simultaneous symbiotic relationship with cyanobacteria and green microalgae, which represent interesting models to analyse the strategies and adaptations resulting from the interactions of lichen symbionts. The present study was designed to contribute to a better understanding of this taxon, which has shown a clear decline over the last century. The symbionts were identified by molecular analysis. The phycobiont is *Symbiochloris reticulata,* and the cyanobionts (*Nostoc*) are embedded in internal cephalodia. Light, transmission electron and low-temperature scanning microscopy techniques were used to investigate the thallus anatomy, ultrastructure of microalgae and ontogeny of pycnidia and cephalodia. The thalli are very similar to its closest relative, *Ricasolia quercizans*. The cellular ultrastructure of *S. reticulata* by TEM is provided. Non-photosynthetic bacteria located outside the upper cortex are introduced through migratory channels into the subcortical zone by the splitting of fungal hyphae. Cephalodia were very abundant, but never as external photosymbiodemes.

## 1. Introduction

Lichens represent major radiations of ascomycetes in a symbiotic stage (lichen-forming fungi) that are characterised by a unique symbiogenetic phenotype of specific biological organisation in the lichen thallus [1,2]. Lichen thalli are complex symbiotic systems, new entities (holobionts), which are individualised from cyclical symbiotic associations [3,4,5]. At least one heterotrophic fungus (mycobiont) and one or more photosynthetic partners (photobionts) that can be cyanobacteria, green microalgae or both make them up. Nevertheless, lichen symbioses have been shown to be far more complex and may include a wide range of other interacting organisms, including non-photosynthetic bacteria, accessory fungi and microalgae [6,7,8]. The cyclical integration of the involved symbionts provides the potential for new and distinct relationships between organisms and generates new “holobionts”: new entities with emergent properties [9]. The relationship between synthrophic metabolism and morphogenesis in the emergence of novelties through physical association is made obvious in their thalli, and so too the contribution of symbiogenesis to speciation and taxonomy [2,4]. In summary, lichens exemplify the details of complex individuality [10].

In some groups, the thallus-forming mycobiont may have a simultaneous symbiotic relationship with cyanobacteria (prokaryotic) and/or green algae (eukaryotic), and namely, tripartite thalli, generally referred to as “photosymbiodemes”, which include lichens forming different thalli, “photomorphs”, with either “phycobionts” or “cyanobionts”. In other thalli, usually foliose, the dominant microalga is located in a layer directly below the cortex in the upper zone of the medulla, while the cyanobacterial cells are confined to some specialised structures, “cephalodia”, which are originated by the mycobiont and may occur in different parts of the thalli [6].

Nylander first categorised cephalodia as epigenic, hypogenic or endogenic, depending on where they originated in the thallus. However, Winter [11] and Forsell [12] appreciated that cephalodia, which form in the lower cortex, can grow into the thallus. Later works refer to the cephalodium simply as internal, if located in the medulla, or external, if located in the upper or lower cortex [13,14,15]. Moreau [16] and Kaule [17] studied the cephalodia of Peltigeraceae (formerly Lobariaceae), indicating that hyphae from the upper or lower cortex grow into, and expand, between adjacent cyanobacterial colonies, enveloping them. In contrast, Jordan [13] found that the cephalodia of the genus *Lobaria* could only begin their formation in the lower cortex, considering that the external cephalodia located in the upper cortex had developed from structures originating from the lower cortex. Nonetheless, Cornejo and Scheidegger [15] indicate that in *L. pulmonaria*, cyanobacterial entry can occur through both the lower and upper cortices.

The family Peltigeraceae (order Peltigerales) comprises many lichens with tripartite thalli, allowing them to cope with changing conditions in their environment [18,19,20]. The lichen we studied in this work, *Ricasolia virens* (With.) H.H. Blom & Tønsberg (=*Lobaria virens* (With.) J.R. Laundon, has recently been included in this family [21]. *Ricasolia virens* is mainly distributed in Western Europe and Macaronesia, in nemoral, well-structured and boreal forests with ecological continuity, as it is sensitive to environmental changes [21,22,23]. It is considered to be a highly vulnerable, endangered or extinct species on the different Red Lists (IUCN) of countries such as the United Kingdom, Sweden, Germany [24], Switzerland [25], Italy [26] and France [27]. It has also experienced a clear decline over the last century [28].

In *R. virens*, the primary phycobiont is *Symbiochloris reticulata* (Tschermak-Woess) Skaloud, Friedl, A. Beck & Dal Grande [29], while the cyanobionts (*Nostoc*) are embedded in small internal cephalodia [30,31], similar to the North American taxon *Ricasolia* (*Lobaria*) *quercizans* (Michx.) Stizenb. [13,32]. *R. virens* is a well-known taxon, although the cephalodia are not referenced in its original description, nor in the typification by Laundon [33], nor in the monographic work of the Spanish and French Lobariaceae [27]. Schumm [31] likewise include the presence of cephalodia in their descriptions, albeit superficially. However, Pérez-Ortega and Barreno [34] found abundant cephalodia in specimens freshly collected on the banks of the Tablizas River (Muniellos MAB Reserve, Asturias, Spain) and highlight their presence as a novelty, as it was the only species of the genus in which their occurrence had not been mentioned. The cyanomorph and photosymbiodemes are reported here for the first time for *Ricasolia virens* (With.) H.H. Blom & Tønsberg comb. nov. (≡*Lobaria virens* (With.) J.R. Laundon).

Recently, Tonsberg et al. [21] reported, for the first time, in various parts of western and central Norway, that *R. virens* develops a new type of dendriscocauloid cyanomorph. They observed early developmental stages involving (1) a free-living cyanomorph; and (2) a photosymbiodeme composed of the cyanomorph supporting small foliose chloromorphic lobes. Whereas the chloromorph continues to grow, the cyanomorph decays and disappears, leading to the final stage, the free-living chloromorph. In *Lobaria pulmonaria*, it has been described that sometimes additional external cephalodia may occur [15,35].

Over recent years, lichen thalli have been shown to be micro-ecosystems, harbouring abundant and diverse bacterial (bacteriobiont) and yeast communities [5,20,36,37,38]. These communities can form biofilm-like structures in specific parts of the lichen thalli [39], and can contribute multiple functions to the lichen symbiotic system [19,40,41,42]. Several authors [43,44,45] have reviewed the structure and composition of non-photosynthetic bacterial communities associated with lichens. More than 800 types of bacteria may contribute to the bacterial microbiome of a single lichen individual [8].

In a former study [30], we found many cephalodia inside the thalli of *R. virens* collected from forests of Community Importance in a Special Area of Conservation, the Sierra del Sueve mountains (Natura 2000-ZEC ES1200043, Asturias, Spain) (BOPA, nº 295). However, we did not find relevant information on the ultrastructure of this lichen, nor on the molecular identification of its symbionts. Thus, we thought it would be interesting to study the anatomy, ultrastructure and ontogeny of the cephalodia, as well as phylogenenetically identify the three symbionts. The ultrastructure of the *Nostoc* that form the cephalodia in these thalli and their interactions were also unknown. Therefore, the aim of this work was to study their anatomy and ultrastructure, as well as the ontogeny of the cephalodia and pycnidia. Moreover, we sought to compare our results with the detailed study (LM) of the North American species *Ricasolia quercizans* carried out by Jordan [13], which he considered to be the closest and a vicariant taxon. In addition, molecular markers and phylogenetic trees have been used to support the identification of the mycobionts, microalgae and cyanobacteria.

An unexpected ultrastructural result was found: non-photosynthetic bacteria were not only present in the lower cortex, but also in the upper cortex, and some of them penetrated and established themselves in niches within the cortex itself. The hyphae of the upper cortex show a similar behaviour to those of the lower cortex, breaking up the plectenchyma in some small areas to originate a kind of corridor that allows very diverse bacteria to pass into the inner parts of the thallus.

## 2. Materials and Methods

### 2.1. Specimens Used for Anatomical and Molecular Studies

Thalli from several populations of the lichen-forming fungus *Ricasolia virens* (*Lobaria virens*) were collected in (a) the Landscape area ZEC (Zona de Especial Conservación, Red Natura, 2000) of the Sierra del Sueve Biescona area (Caravia, Colunga, Asturias) epiphytes on *Fagus sylvatica* L., *Quercus robur* L., *Ulmus glabra* Huds., *Hedera helix* L. and *Castanea sativa* Miller; (b) Vega de Sebarga (banks of the Melón River) on *Fraxinus excelsior*; and (c) Pesoz (banks of the Ahío river) on *Acer pseudoplatanus*, *Quercus robur* and *Hedera helix* (Appendix A). The specimens were dried under ambient conditions, protected from direct light and, before 72 h, stored and kept at a low temperature (−20 °C) in the Lichenology laboratory of the University of Valencia, where the samples were properly preserved [46,47] in order to carry out molecular and anatomical studies, and for the isolation of the phycobionts. Subsequently, some samples were included in the VAL-Lich herbarium, and duplicates were sent to the FCO (Oviedo Science Faculty Herbarium). All the specimens were studied from fresh material, not from herbarium sheets.

### 2.2. Revised Samples in Different Herbaria

The lichen collection of the Herbarium of the Universitat de València, VAL-Lich, was revised (Appendix A). The herbaria MACB (Department of Biodiversity, Ecology and Evolution of the Faculty of Biological Sciences) and MAF-LICH (Department of Pharmacology, Pharmacognosy and Botany of the Faculty of Pharmacy), both in the Complutense University of Madrid, and the MA (Herbarium of the Royal Botanical Garden of Madrid), were also checked.

### 2.3. Molecular Studies

Two individuals for each population were analysed. The samples were dried for one day and then stored at −20 °C until their processing. Lichen thalli were examined under a stereomicroscope to remove soil particles and were immersed sequentially in ethanol and NaOCl [48] to remove surface contaminants and to ensure the intrathalline origin of the sequenced microalgae. Fragments from different parts of the thalli (apical, middle and basal zones) were randomly excised and pooled together [49].

#### 2.3.1. DNA Extraction, Amplification and Sequencing

Total genomic DNA from the mix was isolated and purified using the DNeasy Plant Minikit (Qiagen, Hilden, Germany) following the manufacturer’s instructions. Two algal loci were amplified: the LSU ribosomal RNA gene (rDNA) in the chloroplast, using the algal-specific primers 23SU1 (5′-GGGTAAAGCACTGTTTCGG-3′ (19-mer)) and 23SU2(5′-CCTTCTCCCGAAGTTACGG-3′ (19-mer)) [50]; and nrITS DNA using the primer pair nr-SSU-1780 (5′-CTGCGGAAGGATCATTGATTC-3′ (21-mer)) [51] and ITS4(5′-TCCTCCGCTTATTGATATGC-3′ (20-mer)) [52]. In turn, fungal nrITS DNA was amplified using the primer pair ITS1F (5′-CTTGGTCATTTAGAGGAAGTAA-3′ (22-mer)) [53] and ITS4 (5′-GGCYRWAWCTGACACTSAGGGA-3′ (22-mer)) [52]. To perform cyanobacterial identification, the bacteria-specific 16S rRNA gene was amplified with the cyanobacteria-specific primer pairs 740F (5′-GGCYRWAWCTGACACTSAGGGA-3′ (22-mer)) [54] and 1494R (5′-TACGGCTACCTTGTTACGAC-3′ (20-mer)) [54,55].

PCR reactions were performed following Molins et al. [47]. The PCR products were visualised on 2% agarose gels and purified using the Gel Band Purification Kit (GE Healthcare Life Science, Buckinghamshire, UK). The amplified PCR products were sequenced with an ABI 3100 Genetic Analyzer using the ABI BigDyeTM Terminator Cycle Sequencing Ready Reaction Kit (Applied Biosystems, Foster City, CA, USA).

#### 2.3.2. Phylogenetic Analyses

Three multiple alignments were built. The first was the fungal aligned ITS rDNA with selected sequences of *Ricasolia* sp. pl., *Lobaria* sp. pl. and *Lobarina* sp. pl. from the GenBank. *Dendriscosticta phyllidiata* (MT590923) was included as the outgroup. The second was the algal aligned ITS rDNA with selected sequences of *Symbiochloris* from the GenBank. *Trebouxia jamesii* (FJ626733) was included as the outgroup. The third was considered the cyanobacterial 16S alignment, together with a selection of different *Nostoc* sp. pl. detected in *Pannaria* sp. pl., *Collema* sp. pl., *Peltigera* sp. pl., *Lobaria* sp. pl. and others from undetermined sources, available in the GenBank. *Scytonema singhii* (KT935473) was included as the outgroup.

A multiple alignment was built in MAFFT v 7.0 [56] using default parameters. To determine the evolutionary model that best fit the dataset, we used the program jModelTest v 2.1.4 [57]. Taking into consideration the result of this test, the best model was selected by the Akaike Information Criterion [58] for *R. virens* mycobiont ITS rDNA: K80 + G; for *R. virens* phycobiont ITS rDNA: GTR + G; for Cyanobacterial 16S: GTR + I + G. The phylogenetic relationships were estimated using the Bayesian Inference (BI) and Maximum Likelihood (ML) approaches. ML analysis was implemented in RAxML v 8 [59] using the GTR–GAMMA substitution model. Bootstrap support was calculated based on 1000 pseudoreplicates [60]. BI was carried out in MrBAYES v 3.2 [61]. Settings included two parallel runs with six chains over 20 million generations, starting with a random tree and sampling after every 200th step. We discarded the first 25% of the data as burn-in, and the corresponding posterior probabilities (PPs) were calculated from the remaining trees. The phylogenetic tree was visualised in FIGTREE v 1.4.2 [62,63] (http://tree.bio.ed.ac.uk/software/figtree/ (accessed on 26 January 2023). All analyses were run on the CIPRES Science Gateway v 3.3 web portal [63].

Sequences were deposited under the GenBank accession numbers (Appendix A), and some were also retrieved from the GenBank.

### 2.4. Anatomical and Ultrastructural Analyses

To study the anatomy, development stages and ultrastructural traits of the symbionts of the lichen-forming fungus *Ricasolia virens*, different microscopy techniques were applied.

#### 2.4.1. Fixing of Samples

Several fragments of different thalli containing cephalodia were selected to be observed under light microscopy (LM), scanning electron microscopy (SEM) and low-temperature scanning microscopy (LTSEM). These samples were fixed in a formaldehyde–acetic acid–ethyl alcohol (FAA) mixture and subsequently stored refrigerated until use [64]. Samples for TEM were fixed and dehydrated as described in Molins et al. [65].

#### 2.4.2. Observations under Light Microscopy

The samples fixed with FAA were subjected to the following treatments: (a) a part of the samples was embedded in paraffin [66]; (b) another part of the samples was embedded in LR-White^®^ Hard Grade resin (London Resin Company, London, UK), according to Barreno et al. [30]; (c) finally, other samples were saved for observation using a low-temperature scanning microscope (LTSEM).

All the sections were observed with an OLYMPUS Provis AX 70 (Olympus, Hamburg, Germany) light microscope equipped with epifluorescence equipment. The observations with fluorescence and autofluorescence were performed with a UMWU fluorescence cube (excitation filter: 330–385 nm; barrier filter: 420 nm; dichroic mirror: 400 nm). All the images were photographed with a Lumenera Infinity 2-3C digital CCD colour camera (2080 × 1536 resolution, Lumenera, Ottawa, ON, Canada), captured with the Infinity Analyze^®^ 7 v program. 7.1.0, and subsequently processed with the Photoshop CC^®^ 2018 program, at the Jardí Botànic of the University of Valencia.

#### 2.4.3. Observations under TEM

The samples, fixed with Karnovsky and post-fixed with OsO_4_, were dehydrated in an ethanol series (50%, 70%, 80%, 96% and absolute) for 2 × 15 min each, and then embedded in White^®^ Hard Grade LR-resin (London Resin Company) following the same procedure as for semi-fine sections. Ultrathin sections (80 nm) were cut and viewed according to the procedure outlined in Molins et al. [65]. The ultrathin sections were observed with a JEOL JEM-1010 (80 kV) electron microscope, equipped with a MegaView III digital camera and “AnalySIS” image acquisition software (SCSIE, Universitat de València, València, Spain).

#### 2.4.4. Observations under LTSEM (Cryo-SEM)

Some samples fixed in FAA were used for observation using a low-temperature scanning microscope (LTSEM); the hydrated samples were deposited on a special slide and frozen with liquid nitrogen. Subsequently, they were sublimated at −90 °C for 10 min. Once introduced into the pre-chamber of the microscope, they were covered with gold for 60 s. Finally, they were visualised under a voltage of 1.5 kV on a Field Emission Scanning Electron Microscope (FESEM) (Zeiss Ultra55, Jena, Germany) interconnected to a cryotransfer system (Cryotrans CT1500, Oxford Instruments, Oxford, UK) and equipped with a digital image acquisition system, INCA-Point & ID (Oxford Instruments), at the Electron Microscopy Service of the Polytechnic University of Valencia.

## 3. Results

### 3.1. Phycobiont Identification

The identities of *Symbiochloris reticulata* (Tschermak-Woess) Škaloud, Friedl, A. Beck & Dal Grande as the primary phycobionts were confirmed by two genome markers (LSU rRNA and nrITS DNA). Primers 23SU1 and 23SU2 were used, which showed a unique and clean sequence and, as we specified, are selective for microalgal chloroplasts. This primer pair does not amplify cyanobacteria [50] (Appendix A).

Significant matches of 100–99.85% identity and 95–100% coverage were obtained, respectively, with the *S. reticulata* strain SAG 53.87 (GU017650-GU053573). The aligned phycobionts LSU rRNA and nrITS DNA were 744 and 740 bp, respectively. All the sequences from nrITS DNA formed a well-supported clade (94/100) with the *S. reticulata* sequences included in this analysis. The BI and ML phylogenetic hypotheses were topologically congruent (Appendix A).

### 3.2. Mycobiont Identification

The identities of *Ricasolia virens* (With.) H.H. Blom & Tønsberg mycobionts were confirmed by BLAST analyses against the GenBank database. Significant matches of 99% identity and 100% coverage were obtained with *R. virens* from the United Kingdom (KX385135-KX385141) detected by Cornejo et al. [67], and from Norway, detected by Simon et al. [68]. The aligned fungal ITS was 306 bp. All the sequences formed a well-supported clade (93/100) with the *R. virens* sequences included in this analysis (Appendix A). The BI and ML phylogenetic hypotheses were topologically congruent.

### 3.3. Cyanobacterial Identification

The identity of *Nostoc* sp. was confirmed by BLAST analyses against the GenBank database. From the six lichen thalli analysed, two different *Nostoc* spp. were detected. *Nostoc* sp. 1 was found in the two thalli from the population of VE and the thallus H10. According to GenBank data, *Nostoc* sp. 1 was previously observed in lichens such as *Pannaria* sp., *Peltigera* sp. and *Ricasolia* (*Lobaria*) (Appendix A). *Nostoc* sp. 2 was found in the two thalli from the population of VI and thallus H4. This strain has not been previously detected, though BLAST identification results in 99.45% identity and 100% coverage with *Nostoc* sp. from *Collema* sp. (KF359719) and *Pannaria spinchina* (EF174207). Nevertheless, phylogeny assigns *Nostoc* sp. 2 to a clearly different clade. The aligned partial small subunit of the ribosomal DNA (16S rDNA) was 544 bp. The BI and ML phylogenetic hypotheses were topologically congruent (Appendix A).

### 3.4. Anatomical and Ultrastructural Studies

The macroscopic characteristics of the thalli are extensively described in [31] and Barreno et al. [30] (Appendix A). From the histological point of view, the upper cortex is paraplectenchymateous and well organized (Figure 1A), with the hyphae strongly adhered, which may contain some protoplasts or be empty. Underneath is a thick layer of primary phycobionts, green microalgae of *Symbiochloris reticulata* [23,29] (Figure 1B), organised in about 6–8 rows (Figure 1A and Appendix A) and measuring 4.5–5 μm (Figure 1B,C). The cell walls of these microalgae are thin (120–200 nm). The protoplast has an irregular shape and shows a secretion space, which is clearly visible using TEM (Figure 1B–D). On the periphery of the protoplast there are numerous electrodense vesicles, mostly 150–200 nm in size, although some larger ones also appear (500–600 nm). The chloroplast is plurilobed with highly ordered thylakoid membranes, and among them, small grains of starch can be found, but pyrenoids were not observed, although several pyrenoglobules appear grouped in different areas of the cells. Mitochondria are very scarce (Figure 1C,D).

The medulla is made up of a set of loosely arranged hyphae, arachnoid plectenchyma, with empty spaces remaining (Figure 2A). The lower cortex is also paraplectenchymateous, although sometimes it is disorganized, becoming thinner and fibrous, or it almost disappears. Some of the cells of the lower cortex elongate and originate an abundant tomentum or true rhizomes (Figure 2B and Appendix A). Cephalodia begin their development on the underside, between the hyphae of the tomentum, and then they organize themselves inside the thallus, where they are usually common (Appendix A). The young cephalodia are globose with chains of more or less organized *Nostoc* cells inside and surrounded by a sheath of dense-walled hyphae, which can be easily distinguished from those of the medulla (Figure 2C, Appendix A). The mature ones have cyanobacteria grouped in bundles as a result of the separation made by several threads of hyphae that enter and branch inside, giving a cerebriform appearance (Figure 2D).

The structure of this lower cortex is extremely variable within the thallus itself. Sometimes it can be paraplectenchymateous (Figure 3A), but a few microns away, it can be more disorganized (Appendix A). Most cephalodia originate in the paraplectenchymateous areas along with other non-photosynthetic bacteria (Figure 3B–D and Appendix A). In general, the more compact regions of the lower cortex cover the tomentose areas of the thallus, while the more fibrous cortex is limited to the bare parts of the lower surface. At least in some cases, older, more differentiated cells form the innermost layers of the cortex, while cells on the outermost surfaces are thinner-walled, more elongated and more similar to the medullary hyphae from which they emerge. The hyaline filamentous hyphae that form the tomentum can emerge from any viable cell in the inferior cortex, although they are produced more commonly by the outermost cells.

The development of cephalodia follows several stages: (1) a disruption of the lower cortex in contact with cyanobacteria; (2) the proliferation of the fungal mycelium from the lower layers; and (3) the subsequent sheathing of the cyanobacteria by the hyphae (Figure 3C,D and Appendix A).

During the initial phase of the process, there is no appreciable increase in the number of cyanobacterial cells. The physical incorporation of *Nostoc* cells occurs when the fungal mycelium intrusively envelops and pushes them into the medulla. Once inside the thallus, the cyanobacteria multiply and are isolated by a special type of hyphae that separate and insulate them from the rest of the thallus, thus initiating the development of an internal cephalodium. At the same time, new hyphae develop, pushing the cephalodia into the internal parts of the medulla (Figure 4A–C and Appendix A).

The cephalodia become bounded by a dense layer of fungal tissue, the peripheral sheath (Figure 4B). As cephalodium development progresses, the thickness of the lichen thallus increases (Figure 4B,C) and, within the cephalodia, the cyanobacterium groups appear to be separated from each other by threads of hyphae, with which some cyanobacteria establish contact (Figure 2D, Figure 4D and Appendix A). These tightly packed threads of hyphae that dissect the cephalodia appear to be extensions of the peripheral sheath. The hyphae branching from these threads penetrate the cyanobacterial groups and come into contact with some of them (Figure 4D and Appendix A).

The *Nostoc* cells in symbiosis show ultrastructural differences with respect to the initial ones, which were located externally. Their shape is irregular, with their longest dimension being 3–10 μm (Figure 5A–D). Heterocysts are occasionally observed (Figure 5A,B). In some areas, direct contact can be seen between hyphae and cyanobacteria, but without fungal haustoria, as well as between the heterocysts and surrounding mycobiont hyphae (Figure 5B–D).

Cyanobacteria continue to multiply, resulting in an increase in the size of the cephalodia, which simultaneously move upward to the upper cortex (Figure 5E). The growth of cephalodia can interrupt the layer of green microalgae (Figure 5E). Finally, the cephalodia may appear as a small bump on the upper surface ([30]; p. 353). In these cases, the phycobiont layer may be completely excluded from the area immediately above the cephalodia (Figure 5E,F). Direct contact between microalgae and cyanobacteria has never been observed. Larger cephalodia have ostioles on their surfaces that may be depressed or protruding.

In addition to the cephalodia, in all the samples studied, we found the abundant presence of well-defined pycnidia (330 ± 70 μm), which, at the end of their development, appreciably deform the upper face of the thallus (Appendix A). They generate rod-shaped pycnidiospores (2.9 ± 0.2 × 1.1 ± 0.2 μm) (Appendix A). There are numerous flat to slightly concave apothecia, clearly protruding from the thallus, which are 2–4 mm wide, narrowed at the base and almost stipitated with an orange-pink or intense salmon-coloured disc (Appendix A), being of the zeorin type with the proper excipulum and thalline margin well delimited (Appendix A). Hyaline fusiform, or somewhat yellowish, spores usually have only one septum.

TEM observations show that, in the peripheral zone of the pycnidia near the upper cortex, mycobiont hyphae, arranged in palisade plectenchyma, separate bacterial clusters. In these areas, it can be observed how a direct relationship between the hyphae and the non-photosynthetic bacteria is established, thanks to a matrix of dense content near the hyphae that is vacuolized in the vicinity of the bacteria (Figure 6A,B). The bacteria that appear associated with these hyphae are diverse, showing bacillary forms (1–1.5 × 0.5 μm), with highly developed nucleoids and numerous storage structures, which are dense to electrons; moreover, many of them are fimbriated (Figure 6C). The formation of these surprising and unknown structures begins when bacteria located on the outside of the upper cortex enter through migratory channels to the subcortical zones (Figure 6C–E). These channels appear to originate by the splitting of the cortical hyphae that eventually develop areas or niches where these bacteria accumulate (Figure 6E), and in which they are surrounded by a gelatinous matrix.

## 4. Discussion

Tripartite lichen thalli represent interesting models to analyse the different strategies and adaptations resulting from the interactions of the lichen symbionts [69]. In *Lobaria pulmonaria*, a coexistence consisting of a mycobiont, photosynthetic green alga (*Symbiochloris reticulata*) and *Nostoc* cyanobiont is well known [13,29,32,67,68,70].

Two very different and not phylogenetically related photobionts (green microalgae and cyanobacteria) are able to build two pronouncedly different morphologies by sharing the same mycobiont [69,71], which show symbiotic-specific responses to environmental variables and symbiont-specific optimal ecological conditions [72]. Some of the most sensitive lichens to anthropogenic activities are triple symbiotic systems [73]; thus, knowledge of their biology and adaptations to environmental stresses is truly important for conservation practices [74,75].

The present study was designed to contribute to a better understanding, from a structural and molecular point of view, of the *Ricasolia virens* lichen with tripartite thalli, which is clearly threatened or extinct in many European territories [21,22,23,24,25,26,27,28,76]. Despite its biological and ecological relevance, the only studies on the anatomy and ontogenetic development of the thalli of this lichen were carried out by Letrouit-Galinou [77,78], who particularly analysed the pycnidia or the formation of the plectenchymata of the thalli, although she never mentioned the presence of cephalodia. We also sought to compare our results with the fine anatomical study carried out by Jordan [13,79] on the phylogenetically closest and vicariant species in North America, *Ricasolia quercizans.*

Fungal selectivity for cyanobionts may be strong, as shown by diverse studies [80,81,82,83,84], whereas other studies indicate that selectivity can be low when coexisting in a variable range of lichen species and communities [82,85]. In this regard, the genus *Ricasolia* proved to be able to associate with a wide range of *Nostoc* sp. pl., denoting high flexibility in cyanobiont coexistence [86]. Based on the analysis of 16S rRNA gene sequences, each of the samples analysed from the population in Río Ahío (Appendix A) shows two different cyanobionts, *Nostoc* sp. 1 and *Nostoc* sp. 2, but we did not detect the possible coexistence of both strains in one thallus. Nevertheless, phylogeny assigns *Nostoc* sp. 2 to a clearly different clade. These results indicate that metabarcoding techniques should be used to properly characterise the diversity of the cyanobionts and microbiomes coexisting within this lichen. HTS techniques detect a vast number of genotypes undetectable by conventional PCR amplifications [87,88].

The identification of our samples, using the marker nITS, confirms that the mycobionts are *R. virens* (Appendix A), related to the European populations [67]. Cornejo et al. [67], in their study on populations of *Ricasolia amplissima* in Europe, North America and North Africa, carried out a molecular phylogenetic analysis with three markers, nITS, nRPB2 and mSSU, where they incorporated data from *R. virens* and *R. quercizans* that support the differentiation of these two species. Both the ML and Bayesian analyses showed that they are closely related, but revealed a clear disjunction between the European and eastern North American populations.

The thalli of *R. virens* analysed in this work, as well as those included in the GenBank databases, are associated (pairs up) only with *Symbiochloris reticulata* (Appendix A), which is also found as the sole primary phycobiont in *Lobaria pulmonaria* or *Ricasolia quercizans* [75], which suggests that, as in both these taxa, *R. virens* would appear to be a highly specific symbiosis [23,89,90]. However, in most lichens, a single lichen-forming fungus can be associated with several green microalgae, either in a single thallus or in the same locality, or in different territories [49,88]. Moreover, in the literature, it is well documented that a single microalga species can be associated with several different lichen-forming fungi [91,92,93]. It is noteworthy that mycobionts of the genera *Ricasolia* and *Lobaria* (Peltigeraceae) only pair up with a single species of green microalgae; thus, these are very specific symbioses [91,92,93], suggesting that little flexibility in phycobiont selection by mycobionts of these genera is a noteworthy feature of their biology. The most common scenario is that the same lichen-forming fungus can be paired with diverse species of microalgae, either in different habitats or different biogeographic areas, and there are numerous examples of this in the literature; it is also not uncommon to find cases of the coexistence of diverse species and/or genera in the same thallus ([94], and citations therein). Conversely, when the thalli of *Lobaria pulmonaria* and *Ricasolia* sp. pl. share the same habitat, they also share a unique phycobiont, *S. reticulata*, with genetic structures that are very similar among the two symbionts [92,93,95].

In this work, as a novelty, we provide the cellular ultrastructure of the microalga *Symbiochloris reticulata*, using TEM, which shows a protoplast with an irregular shape and a clearly differentiated secretion space; the chloroplast is plurilobed with highly ordered thylakoid membranes, although Škaloud et al. [29] describe it as being circular when observed by LM. The pyrenoids could not be seen, although the pyrenoglobules appeared grouped in different areas (Figure 1B–D), and the cell walls are thin. We do not know whether the ultrastructure observed in this microalga is similar to those of other *Symbiochloris* species, as there is no graphic information available. According to Dal Grande et al. [92], this microalga, *S. reticulata,* is associated only with different genera of the family Lobariaceae (at present Peltigeraceae) in both nemoral and boreal forests of Europe and in the subtropics. Although Dal Grande et al. [92] studied the molecular phylogeny and species selectivity of this genus in wide territories, and Škaloud et al. [29] made a whole taxonomic revision of the genus, they only provided LM images of its cells.

Allen and Scheidegger [75] tested the photobiont-sharing capacity of *Ricasolia quercizans* with *L. pulmonaria* using nine microsatellite markers as an example of a photobiont-mediated guild of *Symbiochloris reticulata*, where *L. pulmonaria* was considered the core species of this guild, concluding that the populations of this microalga were not significantly genetically distinct. Dal Grande et al. [92], when analysing other species of *Ricasolia* in Europe and Asia, found something similar. Allen and Scheidegger [75] point out that these results imply that there could be a correlation between the continued decline in *L. pulmonaria* in the forests of southeastern North America, and a greater decline in *R. quercizans*, because of the scarce availability of the shared phycobiont. In fact, in Switzerland, the decline in *L. pulmonaria* has resulted in reported declines in *R. amplissima* (Endangered) and *R. virens* (Regionally Extinct) [76]. Consequently, the limited flexibility to select different phycobionts may be a serious problem for the conservation of populations of such tripartite lichens of the family Peltigeraceae, subjected to significant anthropogenic disturbances. In Europe, the phylogeographic structures are highly similar for *L. pulmonaria* and *S. reticulata* [91]. The results obtained with the thalli of the populations of *R. virens* analysed in this work seem to point out that similar models to those cited above could be comparable when populations of *R. virens* with a wider distribution and greater number of specimens are studied.

In short, *R. virens* thalli show plectenchyma in the upper cortex (25–45 µ), with 4–6 cell layers and a thick phycobiont layer (40–45 µm, made up of 7–10 µm cells; the medulla is 130–240 µm thick; the lower cortex is 13–20 µm, and 2–4 cells thick). Pycnidia with black depressed ostioles are abundant. These data are very similar to those described by Jordan [79] for *R. quercizans*, the main differences with *R. virens* being that in *R. quercizans*, the pycnidiophores with pycnidiospores are 1.5–3.5 µm, and the internal cephalodia are scarce and small.

The abundance of cephalodia in all our specimens allowed us to study different stages of their ontogeny. Although Schumm [31] likewise include the presence of cephalodia in their descriptions, albeit superficially, and they were mentioned by Cannon et al. [96], it is likely that they have not been detected by other authors because when the thalli are dehydrated, it is difficult to observe the different stages, together with the fact that they show some resemblance to the abundant primordia of the apothecia and pycnidia on the upper face and, also, due to the dense tomentum on the lower face. Neither in the original description of *Lichen virens* With. (1776) nor in the typification made by Laundon in 1984 [33] was there any mention of the presence of cephalodia. Therefore, in 2007, E. Barreno requested Dr. M.R. Seaward (U.K.) to check for the presence or absence of cephalodia in the type specimen at the Dillenius Herbarium (Oxford Lichen Herbarium). The results of this check were unsuccessful, as it was impossible to observe the underside of the thallus due to the processing of the thallus in the sheet (private communication). Likewise, this abundance of cephalodia has not been clearly detected in the numerous herbarium sheets reviewed (in the VAL-Lich, MACB, MAF or MA), possibly due to the fact that most of the cyanobacterial aggregates that externally initiate the process of migration towards the interior disappear when they are dehydrated for a long time [30]. Therefore, only those developed inside the thalli, which are more difficult to observe, would be better preserved. Accordingly, when dealing with macrolichens with two different types of photobionts, studies should be carried out on the freshest samples available, as was the case in this study.

Moreau [16] and Kaule [17] studied the cephalodia of Lobariaceae, indicating that the hyphae of the upper or lower cortex grow towards the adjacent cyanobacterial colonies and expand between them, enveloping them. In contrast, Jordan [13,79], with his detailed anatomical studies, found that the cephalodia of the genus *Lobaria* can only begin their formation in the lower cortex, considering that the external cephalodia located in the upper cortical zones developed from those originating in the lower cortex. Based on our observations, in *R. virens*, the *Nostoc* colonies that would form the cephalodia do not actively penetrate the host tissue. Their incorporation occurs when they interact with the fungal hyphae. Jordan [13,79] already proposed that the hyphae in contact with *Nostoc* cells could be stimulated in their differentiation by some secretion of the cyanobacteria, which must be quite specific, as the mycobiont does not antigenically respond in the same way to other genera of epiphytic organisms that also inhabit the underside of the thallus. In contrast, Galun and Kardish [97] demonstrated that the recognition of compatible photobiont cells is carried out by specific lectins produced by the mycobiont. More recently, this recognition has been investigated in *Peltigera* species to identify different types of lectins and their involvement in the recognition of symbiotic partners [98,99,100,101,102]. A lectin has been found in *P. aphthosa* that detects *Nostoc*-compatible cells at the onset of cephalodium formation, and this process is highly specific [103,104]. Genes encoding two lectins involved in photobiont recognition have been identified [105]. Likewise, results on gene expression in tripartite thalli under different environmental conditions are starting to be obtained, but for the moment, only under temperature increases [71].

Cornejo and Scheidegger [15] demonstrated that in *L. pulmonaria*, cyanobacteria can be incorporated through both the upper and lower cortices due to the considerable plasticity found in the paraplectenchyma. However, in the case of *R. virens*, plasticity is restricted to the lower cortex, with the upper cortex featuring several layers of thick-walled cells that are highly agglutinated, as occurs in other species of this group [13], which hinder the entry of cyanobacteria, but not of other non-photosynthetic bacteria (Figure 6A–E).

It is relevant to note that we observed evident changes in the morphology of *Nostoc* sp. pl. when they are in a symbiotic state inside the developing cephalodia. The TEM results show that, in the cephalodium, the cyanobacteria present an irregular shape (Figure 5A), which is probably due to the mechanical pressure exerted on them by the hyphae that surround them, as already suggested by Jordan [13,79]. However, the linear arrangement that the cyanobacteria exhibit when they are in the external environment associated with the lower cortex disappears (Figure 3C,D). The presence of haustoria has not been observed, although direct contact between hyphae and *Nostoc* cells was distinguished. The establishment of these contacts is essential for the association, as they allow nutrient transfer between symbionts [71], which is often considered the functional core of lichen symbioses.

According to our observations, once cyanobacterial colonies are introduced into the thallus, they grow and expand through the medullary tissue. These are distributed throughout the thallus, and although the larger ones are more abundant in the older portions of the lichen thallus, young cephalodia can also be found in the vicinity. Mycobionts in triple symbiotic systems are known to exhibit obligate “cyanotrophy” [106,107]. Many of these lichens have large thalli and grow in nutrient-poor habitats. They depend on an efficient nitrogenase activity system that occurs in cyanobiont heterocysts [108,109]. Kershaw and Millbank [110] studied the nitrogen metabolism of *Nostoc* of cephalodia in *Peltigera* and *L. pulmonaria* [111], and they proved that practically all of the nitrogen fixed by *Nostoc* is used by the mycobiont, which was confirmed more recently in *Lobaria oregana* [112], *L. pulmonaria* [113] and *Stereocaulon vesuvianum* [114]. These results could be extrapolated to the thalli of *R. virens*, which could explain the large size of its thalli.

The cyanobacteria of cephalodia have been shown to have higher frequencies of heterocysts and higher rates of nitrogen fixation than those of bipartite cyanolichens [115,116,117]. This nitrogen-fixing capacity of cyanobacteria has two effects on the adaptations to different ecological conditions of lichens. On the one hand, lichen species harbouring N_2_-fixing cyanobacterial symbionts show higher concentrations of organic nitrogen compared to species without cyanobacteria [118]. Accordingly, Büdel and Scheidegger [119] conclude that cyanolichen thalli may have certain advantages in colonising special ecological niches, such as extremely oligotrophic habitats [120,121], and, in turn, the decomposition of these large cyanolichens appears to contribute significantly to the organic nitrogen supply in forest ecosystems [122,123,124,125,126,127,128], which could be the case for *R. virens* in the habitats studied in this work.

It is noteworthy that, as in our study, direct physical contact between cyanobacteria and green microalgae has also not been observed in other tripartite species [85,115,117,129,130,131]. This may be because both free-living and symbiotic cyanobacteria can produce toxins when stressed [117,132]. The effects of these toxins on the fungus and, in the case of tripartite lichens, on microalgae, as well as other components of lichen microbiomes, are still poorly understood [116,133,134].

Phycobionts provide carbohydrates to the mycobiont, which develops the structure and morphology of the thalli, serving as the scaffold of the whole symbiotic system and providing niches for the establishment of communities of other types of non-photosynthetic bacteriobionts [43,135,136], which play a key role in supplying nitrogen to their symbiotic partners [137,138,139]. This could involve both bacteriobionts living inside the thallus, and those developing in parts of the thallus influenced by the mycobiont hyphae, called the “hypotalosphere” by Grube and Berg [43,140], also contributing to the supply of nitrogenous compounds to the thalli.

In this work, we not only detected the presence of these non-photosynthetic bacterial communities in the upper cortex, but we also observed, in certain areas (Figure 6), direct interactions between bacterial clusters and some hyphae of the paraplectenchymateous cortex. The formation of these unexpected and unknown structures begins when bacteria located outside the upper cortex are introduced through migratory channels into the subcortical zone (Figure 6C–E). These channels seem to originate by the splitting of fungal hyphae that eventually develop areas, or niches, where these bacteria accumulate (Figure 6E) and in which they are surrounded by a gelatinous matrix. This fact corresponds with the multi-functionality and considerable plasticity of the upper cortex that form the regenerative structures observed by other authors [141] on *L. pulmonaria* and other species of this genus.

These bacteriobiont communities, which had so far only been observed in the outer zones [43,135,136], may have very different functions: lytic activities, nitrogen fixation and the production of bioactive substances, including hormones and antibiotics [142,143,144]. In many lichens studied, α-proteobacteria form the most abundant and metabolically active bacterial class, and particularly the Rhizobiales [20,36,43,145]. These non-photosynthetic bacteria can colonize different parts of the thallus; α-proteobacteria are widespread on both the upper and lower sides of *L. pulmonaria* [146,147], on *Umbilicaria* sp. [40], in *Ramalina farinacea* [138] and in *Parmotrema pseudotinctorum* [39,139]. In contrast, β-proteobacteria are locally restricted to the underside [40].

The contribution of bacteria to lichen symbioses may not be restricted to a particular function in the thallus system, but rather suggests that it is a complex functional network that still needs to be studied in more detail [43]. Significant research has been carried out on mycobiont–photobiont interactions, although it is evident that, in a single tripartite thallus, the mycobiont responds with different morphological structures to the relationship with each type of photobiont. However, little is known about the control that microalgae and cyanobacteria have on the fungal phenotype [23,91,148,149,150].

In conclusion, the results obtained in this study show relevant novelties in the morphogenetic processes of *R. virens* that highlight the need to analyse the thalli of this lichen using HTS techniques to identify the numerous symbionts that make up this complex microecosystem (holobiome). Moreover, the isolation of bacterial symbionts for laboratory experiments is necessary to understand the adaptations and their role in the functioning of these thalli, as well as their contribution to the morphogenesis and interactions between partners.

In this study, we found abundant populations of *R. virens* in different forests in Asturias (northern Spain), which could be interpreted as “refuge areas” for this lichen in Europe. Nevertheless, we also found some thalli with visible signs of damage in phorophytes close to trails, forest tracks and zones frequented by animals [151], which indicates the presence of disturbances that may threaten their long-term survival in these forest systems.

## Figures and Tables

**Figure 1 microorganisms-11-01517-f001:**
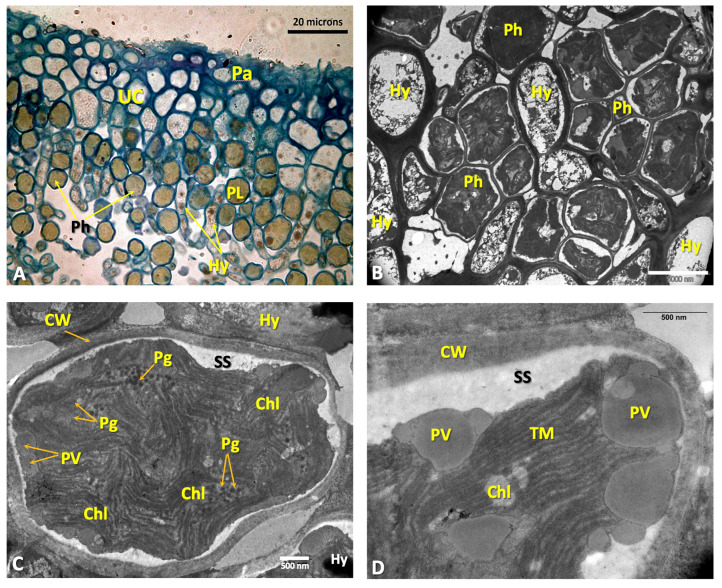
(**A**) LM micrograph of a semi-thin transversal section of the upper cortex of a *Ricasolia virens* thallus stained with toluidine blue with the cortical paraplectenchyma, primary phycobiont layer and medullary hyphae. Bar: 20 microns. (**B**) TEM micrograph showing a detailed view of the phycobiont layer (*Symbiochloris reticulata*). Bar: 2000 nm. (**C**) TEM micrograph showing the internal organization of a phycobiont. The chloroplast structure and the presence and location of peripheral vesicles are visible. Bar: 500 nm. (**D**) TEM micrograph of the inside of a phycobiont cell with the chloroplast and peripheral vesicles. Bar: 500 nm. Abbreviations: CW: cell wall; Chl: chloroplast; Cy: cyanobacteria; Hy: hyphae; LC: lower cortex; M: medulla; Pa: paraplectenchyma; PL: phycobiont layer; Pg: pyrenoglobuli; Ph: phycobiont cells; PV: peripheral vesicles; SS: secretion space; UC: upper cortex.

**Figure 2 microorganisms-11-01517-f002:**
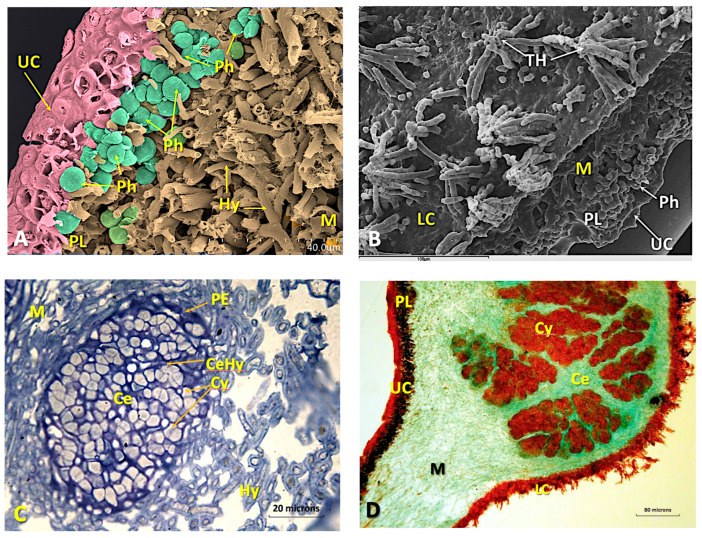
(**A**) False-colour LTSEM micrograph showing a detailed view of the internal structure of a *Ricasolia virens* thallus, the upper cortex with paraplectenchymateous organization, the phycobiont layer and medullary hyphae. Bar: 40 microns. (**B**) LTSEM micrograph showing the tomentose hyphae in the lower cortex and the inner part of the thallus. Bar: 100 microns. (**C**) LM micrograph of a semi-thin transverse section, stained with toluidine blue, in the medullary zone, near the lower cortex, of an *R. virens* thallus where a cephalodium is developing. Bar: 20 microns. (**D**) LM micrograph of a transversal section of a thallus showing a developing cephalodium deforming the upper cortex. Sample embedded in paraffin and stained with safranin/fast green. Bar: 80 microns. Abbreviations: Ce: cephalodium; CeHy: cephalodial hyphae; Chl: chloroplast; Cy: cyanobacteria; Hy: hyphae; LC: lower cortex; M: medulla; PE: peripheral envelope; PL: phycobiont layer; Ph: phycobiont cells; TH: tomentose hyphae; UC: upper cortex.

**Figure 3 microorganisms-11-01517-f003:**
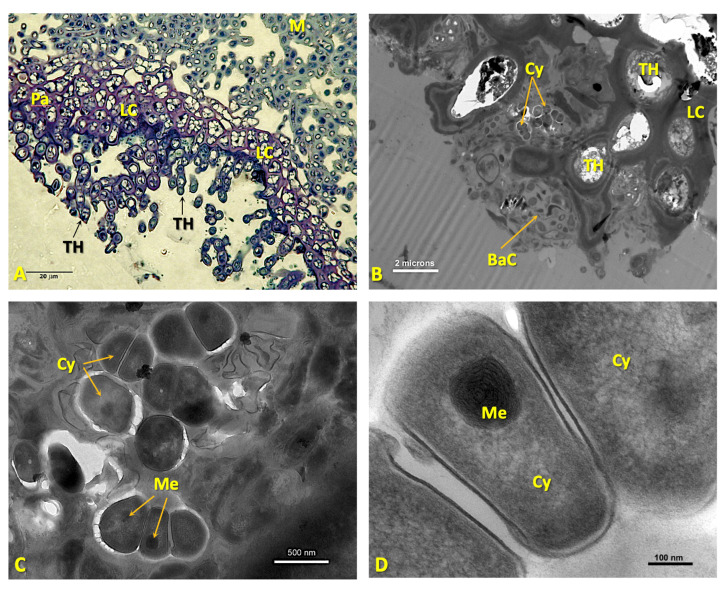
(**A**) LM micrograph of a semi-thin transverse section of the lower cortex of a *Ricasolia virens* thallus, stained with toluidine blue. The tomentose hyphae and paraplectenchymateous organization are apparent. Bar: 20 µm. (**B**) TEM micrograph of the lower cortex showing some colonies of cyanobacteria and non-photosynthetic bacteria affixed to it. Bar: 2 µm. (**C**) TEM micrograph of cyanobacterial colonies adhering to the lower cortex. Bar: 500 nm. (**D**) TEM micrograph of one of the cyanobacteria in the outer colonies adhered to the cortex. Bar: 100 nm. Abbreviations: BaC: non-photosynthetic bacterial colony; Cy: cyanobacteria; Hy: hyphae; LC: lower cortex; M: medulla; Me: mesosome; Pa: paraplectenchyma; TH: tomentose hyphae.

**Figure 4 microorganisms-11-01517-f004:**
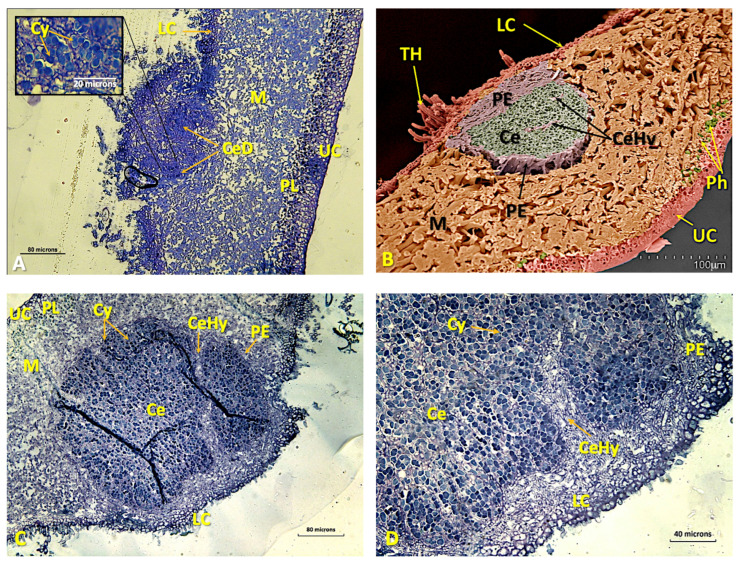
(**A**) LM micrograph of a semi-thin transversal section of the lower cortex of a *Ricasolia virens* thallus, stained with toluidine blue. The entrance of cyanobacteria through the lower cortex and the beginning of the formation of a cephalodium are distinguished. Bar: 80 microns. (**B**) False-colour LTSEM micrograph showing the formation of a cephalodium from the lower cortex. (**C**) LM micrograph of a semi-thin cross-section of the lower cortex and medulla of a *R. virens* thallus, stained with toluidine blue. The growing cephalodium is occupying a large part of the medullary area. Bar: 80 microns. (**D**) LM micrograph of a semi-thin cross-section of a developing cephalodium of *R. virens*, stained with toluidine blue. The presences of cyanobacteria, cephalodial hyphae and peripheral sheath are visible. Bar: 40 microns. Abbreviations: Ce: cephalodium; CeD: cephalodium in development; CeHy: cephalodial hyphae; Cy: cyanobacteria; LC: lower cortex; M: medulla; PE: peripheral sheath; PL: phycobiont layer; Ph: phycobiont cells; TH: tomentose hyphae; UC: upper cortex.

**Figure 5 microorganisms-11-01517-f005:**
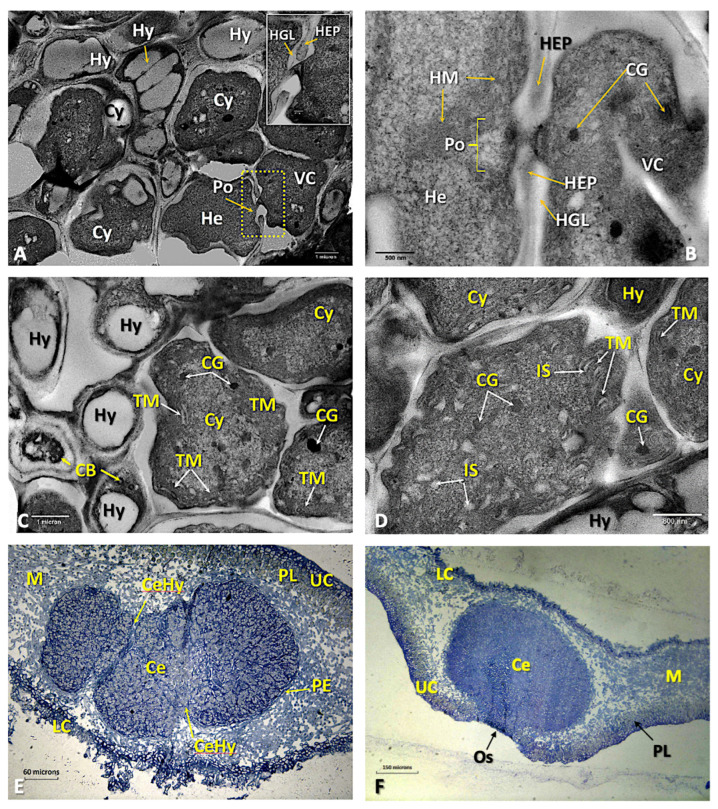
(**A**) A detailed TEM micrograph of a cephalodium of *Ricasolia virens.* The ultrastructure of the cyanobacteria, cephalodial hyphae, some vegetative cells and hyphae and a heterocyst with its pore are observable. The upper right frame depicts a detail of the pore outlined in dashed yellow. Bar: 1 micron. (**B**) A TEM micrograph in fine contrast of a communication pore between a heterocyst and vegetative cell. Bar: 500 nm. (**C**) TEM micrograph showing the interaction between cephalodial hyphae and cyanobacteria. Bar: 1 micron. (**D**) TEM micrograph showing the internal structure of a cyanobacterium. Bar: 800 nm. (**E**) LM micrograph of a semi-thin cross-section of a developing cephalodium, stained with toluidine blue. The structure of the cephalodium and its central position in the medulla are shown. Bar: 60 microns. (**F**) LM micrograph of a semi-thin cross-section of a mature cephalodium of *R. virens* stained with toluidine blue. The deformation of the thallus can be observed, and an ostiole opening is formed upon contact with the upper cortex. Bar: 150 microns. Abbreviations: Ce: cephalodium; CeHy: cephalodial hyphae; CG: cyanophycin granules; Cy: cyanobacteria; He: heterocyst; HEP: heterocyst polysaccharides; HGL: heterocyst glycolipids; HM: honeycomb membranes; Hy: hyphae; IS: intermembranes space; LC: lower cortex; M: medulla; Os: ostiole; PE: peripheral envelope; PL: phycobiont layer; Po: porum; TM: thylakoid membranes; UC: upper cortex; VC: vegetative cell.

**Figure 6 microorganisms-11-01517-f006:**
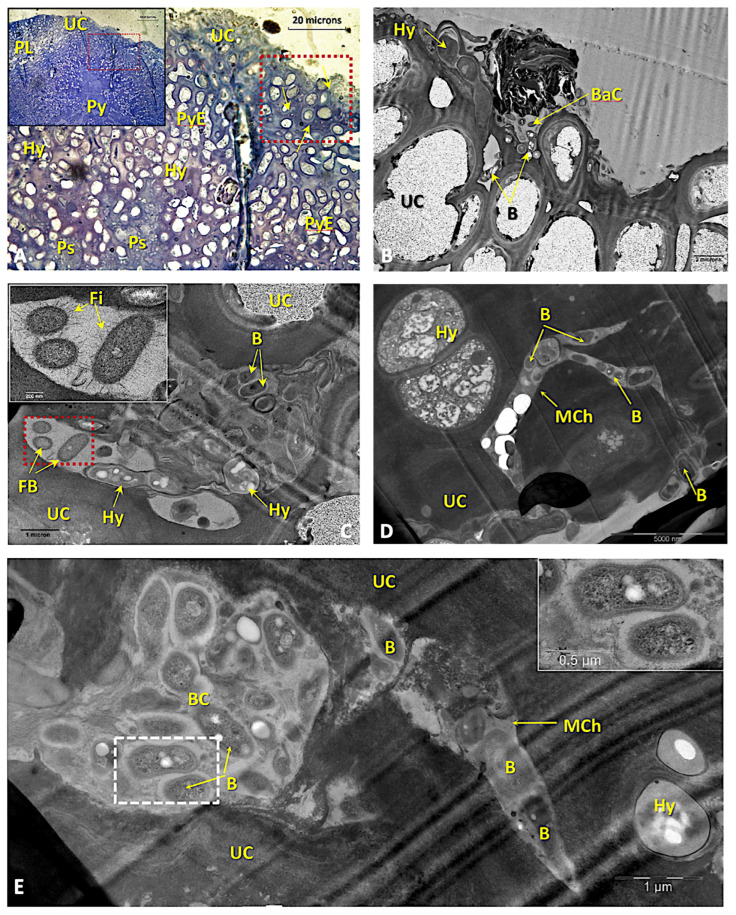
(**A**) LM micrographs of a semi-thin cross-section of the upper cortex of *Ricasolia virens*, stained with toluidine blue. An area with the formation of channels produced by the mycobiont through which some non-photosynthetic bacteria can penetrate and settle inside the thallus is evident. Bar: 20 microns. (**B**) TEM micrograph of the area outlined within the red box in (**A**). Arrangement of epiphytic bacterial colonies on the upper cortex and an entrance channel. Active mycobiont hyphae were always found near these channels adjacent to a pycnidium. Bar: 2 microns. (**C**) TEM micrograph of a zone of non-photosynthetic bacterium accumulation inside the upper cortex with abundant fimbriate bacteria (inset) and active hyphae. Bar: 1 micron. (**D**,**E**) TEM micrographs of the area outlined by a red box in the figure. It shows areas of the upper cortex where the entrance channels of these bacteria appear. The square in (**E**) shows the structure of the non-photosynthetic bacteria found inside the channels. Bar: 5000 nm (**D**); 1 micron (**E**); 0.5 microns (inset (**E**)). Abbreviations: B: non-photosynthetic bacteria; BaC: non-photosynthetic bacterial colony; Fi: bacterial fimbriae Hy: hyphae; M: medulla; MCh: mycobiont channel; PE: peripheral envelope; PL: phycobiont layer; Py: pycnidium; PyE: pycnidium sheat; PyS: pycnidiospores; UC: upper cortex.

## Data Availability

The datasets generated during the current study are available in the GenBank (see Appendix A): ITS Mycobiont: OP925492-OP925496; Photobionts: ITS: OP932022-OP932026; 16 S: OP925747-OP925751; 23SU: OP932028-OP932033.

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
