# Peer review of "The Tripartite Lichen Ricasolia virens: Involvement of Cyanobacteria and Bacteria in Its Morphogenesis"

_microorganisms, 2023, doi:10.3390/microorganisms11061517_

Round 1

Reviewer 1 Report

The authors provide a valuable, detailed perspective into the tripartite lichen Ricasolia virens.  The study was designed well and reported effectively in the manuscript. The authors provide appropriate background for understanding their study in the Introduction section, and explore their results in a meaningful way in the discussion. Apart from a few grammatical and typographical issues, I recommend the study for publication.

Throughout the Results section, there are quite a few cases of hyphenated words that should not be hyphenated - please correct those. I did not carefully consider formatting in the References list and encourage the authors to double check that those are done correctly. Finally, I provide a few specific comments by line below.

14 - is a comma needed here?

15 - consider "extirpated", rather than "extinct"

21 - "analysEs", rather than "analysis"

22 - define acronyms

45 - consider replacing "the details" with "a perspective"

69 - delete "have"

73 - consider "extirpated", rather than "extinct"

104 - add a "," after "[47]"

188 - delete "genome"

234 - italicize "Symbiochloris reticulata"

236 - please correct "match-es"

237 - correct "re-spectively"

239 - "re-spectively."

242 - correct "data-base"

245 - replace "pb." with "bp"

433 - delete "The"

446 - consider "extirpated", rather than "extinct"

471 - replace "fungi" with "fungus"

Author Response

Referee 1

Comments and Suggestions for Authors

The authors provide a valuable, detailed perspective into the tripartite lichen Ricasolia virens. The study was designed well and reported effectively in the manuscript. The authors provide appropriate background for understanding their study in the Introduction section, and explore their results in a meaningful way in the discussion. Apart from a few grammatical and typographical issues, I recommend the study for publication.

Thank you very much for your comments, recommendations and for considering that “The study was designed well and reported effectively in the manuscript, etc.”.

We have checked the grammatical and typographical issues.

Comments on the Quality of English Language

Throughout the Results section, there are quite a few cases of hyphenated words that should not be hyphenated - please correct those. I did not carefully consider formatting in the References list and encourage the authors to double check that those are done correctly. Finally, I provide a few specific comments by line below.

14 - is a comma needed here?

Done.

15 –consider “extirpated” rather than “extinct”

However, extinct” is the terminology used by the UICN categorization of species risks.

21 - "analysEs", rather than "analysis".

In this case, it is a general term for this methodology, not for the amount of analysis performed.

22 - define acronyms.

Indeed, acronyms should be specified the first time they are mentioned and this is how we have done this in the text. The problem here is that the number of words in the Abstract is limited to 200.

We have accepted your suggestion and have removed the term North American for Ricasolia quercizans to include the specification of these acronyms. Now in line 22

45 - consider replacing "the details" with "a perspective"

In this case, we have accurately transcribed Lynn Margulis's original sentence in her book “Symbiosis in Cell Evolution” (10)

69 - delete "have"

Done. Now in line 72

73 - consider "extirpated", rather than "extinct"

However, “extinct” is the terminology used by the UICN categorization of species risks.

104 - add a "," after "[47]"

Done. Now in line 112

188 - delete "genome"

Done. Now in line 198

234 - italicize "Symbiochloris reticulata"

Done. Now in line 246

236 - please correct "match-es"

Done. Now in line 248

237 - correct "re-spectively"

Done. Now in line 249

239 - "re-spectively."

Made by the template. Now in line 250

242 - correct "data-base"

Done. Now in lines 273

245 - replace "pb." with "bp"

Done. Now in lines 276

433 - delete "The"

Done. Now Tripartite lichen in lines 472

446 - consider "extirpated", rather than "extinct"

However, “extinct” is the terminology used by the UICN categorization of species risks.

471 - replace "fungi" with "fungus"

Done. Now in lines 516

Comments on the Quality of English Language

Throughout the Results section, there are quite a few cases of hyphenated words that should not be hyphenated - please correct those. I did not carefully consider formatting in the References list and encourage the authors to double check that those are done correctly.

Thank you very much. We have reviewed the hyphenated words.

The hyphenated words in the text were produced when we included the written texts in Word format within the "template" of the journal, this was done by the automatic space editors. In the final version of the ms. we will carefully check these irregularities in the text. The reference list was revised and emended.

Reviewer 2 Report

A very interesting and complete study is presented. It would be interesting to know the phylogeny of such non-photosynthetic bacteria and to be able to describe their function in the conformation, structure and health of the lichen.

Author Response

Comments and Suggestions for Authors

A very interesting and complete study is presented. It would be interesting to know the

phylogeny of such non-photosynthetic bacteria and to be able to describe their function in the

conformation, structure and health of the lichen.

Thank you very much for your comment “A very interesting and complete study is presented”.

We agree with your comment “It would be interesting to know the phylogeny of such non-photosynthetic bacteria and to be able to describe their function in the conformation, structure and health of the lichen”

As pointed out in the final conclusion (lines765-769): “ In conclusion, the results obtained in this study show relevant novelties in the morphogenetic processes of R. virens that highlight the need to analyse the thalli of this lichen using HTS techniques to identify the numerous symbionts that make up this complex microecosystem (holobiome). Also, the isolation of bacterial symbionts for laboratory experiments is necessary to understand the adaptations and their role in the functioning of these thalli, as well as their contribution to morphogenesis and interactions between partners.”.

Reviewer 3 Report

The ultrastructure of the Ricasolia virens provides important information to understand interactions among mycobiont, photobionts, and other components of the lichen thalli. Especially it was very interesting to show the ultra-structures that are related with non-photobiont bacteria. It is because the lichen thalli can form direct interaction with bacterial species. However, I do not think that the manuscript was well prepared to show the important results of this study. I suggest to revise the manuscript to show the scientific information more clearly.

1. The title “The Ricasolia virens holobiome: involvement of cyanobacteria and bacteria in its morphogenesis” overexpress the contents of the manuscript. The word “holobiome” represents all the organisms in the lichen including mycobiont, photobiont, mycobiome such as lichenicholous fungi, endolichenic fungi, and all the bacteria. However, this study only shows the mycobiont, phycobiont, cyanobiont and a few bacteria located in the upper cortex. In addition, I do not agree authors provide clear evidence for “the involvement of cyanobacteria and bacteria in its morphogenesis”. They showed that some structural features when cyanobacteria and bacteria are included in the inside of the lichen thallus, but did not show the process of morphogenesis. Please choose appropriate expressions to cover the important information of this study.

2. line 19-20: “The present study was designed to contribute to a better understanding of this taxon, showing a clear decline over the last century”. I cannot find the data to show “a clear decline over the last century”. Please describe exactly what you found in this study and focus on the interpretation of the results.

3. Significant match-es of 100% - 99.85% identity and 95% - 236 100% coverage were obtained, re-spectively, with the S. reticulata strain SAG 53.87 237 (GU017650- GU053573). Aligned phycobionts LSU rRNA and nrITS DNA were 744 and 238 740 bp, re-spectively  à Blast search is not enough to determine the identify of the photobionts. Please try phylogenetic analyses with reliable reference sequences and include the tree in supplementary materials like the mycobiont tree.

4. Line 160-161: To perform cyanobacterial identification, the bacterial specific 16S rRNA gene 160 was amplified with the primer pair 740F [56] and 1494R [57]. à Lichen materials contain diverse bacterial species and it is expected that many species of bacterial rRNA genes are amplified. It is not clear how cyanobacterial sequences were selectively amplified or selected from clones or microbiome sequences. Please make it clear how cyanobacterial sequences were selectively obtained.

5. Line 194-197: Several fragments of different thalli containing cephalodia were selected and fixed separately in: (a) a mixture of formaldehyde-acetic acid-ethyl alcohol (FAA) (to be used under light microscopy -LM-, scanning electron microscopy -SEM-, and low-temperature scanning microscopy – LTSEM-) and then stored in a refrigerator à Please make the sentence clearly understandable.

6. Line 251, 253: Strain 1, Strain 2 à The word strain is usually used for an established pure culture. Please use more appropriate expression.

7. Citing references in the middle of the sentences disturb reading the paper. Please include citation at the end of the sentence unless it is very critical to cite in the middle of the sentence. For example,

Line 95: In Lobaria pulmonaria. [15,35] it was described that sometimes additional external cephalodia may occur à In Lobaria pulmonaria, it was described that sometimes additional external cephalodia may occur [15,35]

Line 434-436: In Lobaria pulmonaria, a coexistence consisting of a mycobiont [69,70], a photosynthetic green alga 435 [Symbiochloris reticulata; 29,32] and a Nostoc cyanobiont [13,72] is well known  à In Lobaria pulmonaria, a coexistence consisting of a mycobiont, a photosynthetic green alga (Symbiochloris reticulata) and a Nostoc cyanobiont is well known (13,72, 29,32, 69,70).

8. Typolological errors should be corrected. For example,

Line 234: Symbiochloris reticulata à italic

Line 236: Significant match-es à Significant matches

Line 237, 239: re-spectively à respectively

Line 237: S. reticulata à italic

Line 242: data-base à database

Line 243: ob-tained à obtained

Line 244: de-tected à detected

Line 257: Strain 2 à strain 2

9. Discussion parts are majorly composed of data from other studies and only very little is to interpret the results of this study. à Please focus on interpretation of the results of this study.

Author Response

Referee 3:

Comments and Suggestions for Authors

The ultrastructure of the Ricasolia virens provides important information to understand interactions among mycobiont, photobionts, and other components of the lichen thalli. Especially it was very interesting to show the ultra-structures that are related with non- photobiont bacteria. It is because the lichen thalli can form direct interaction with bacterial species. However, I do not think that the manuscript was well prepared to show the important results of this study. I suggest to revise the manuscript to show the scientific information more clearly.

Thank you very much for your comments and for considering that this study provides important scientific information to understand interactions among mycobiont, photobionts, and other components of the lichen thalli.

We have revised the manuscript following your suggestions.

  1. The title “The Ricasolia virens holobiome: involvement of cyanobacteria and bacteria in its

morphogenesis” overexpress the contents of the manuscript. The word “holobiome” represents all the organisms in the lichen including mycobiont, photobiont, mycobiome such as lichenicholous fungi, endolichenic fungi, and all the bacteria. However, this study only shows the mycobiont, phycobiont, cyanobiont and a few bacteria located in the upper cortex. In addition, I do not agree authors provide clear evidence for “the involvement of cyanobacteria and bacteria in its morphogenesis”. They showed that some structural features when cyanobacteria and bacteria are included in the inside of the lichen thallus, but did not show the process of morphogenesis. Please choose appropriate expressions to cover the important information of this study.

We have accepted the change of "Holobiome" since we have only studied a part of the holobiome of R. virens. In fact, this term is quite widespread in lichenological literature and has been imprecisely used. For this reason, we have changed the title to "The tripartite lichen".

The new title is:

“The tripartite lichen Ricasolia virens: involvement of cyanobacteria and bacteria in its morphogenesis”

However, we would like to make several considerations about the conceptual differences in morphogenetic processes. According to one of the latest analyses (2021) carried out by authors who study these processes in plants": Marco Marconi & Krzysztof Wabnik: Shaping the Organ: A Biologist Guide to Quantitative Models of Plant Morphogenesis. “Organ morphogenesis is the process of shape acquisition initiated with a small reservoir of undifferentiated cells. In plants, morphogenesis is a complex endeavour that comprises a large number of interacting elements, including mechanical stimuli, biochemical signalling, and genetic prerequisites”….”. Based on Thompson (1917): was that morphogenesis could be summarized as a series of coherent geometrical transformations leading to the spacious diversity of biological forms. The concept of morphogenesis is therefore quite general; we usually define morphogenesis as a recipe to build an organism with elements such as individual cells, genes products and biochemical signals. While many cells proliferate to recreate the organism’s adult shape some of them may differentiate into specialized tissues”.

We think that we provide clear evidence for “the involvement of cyanobacteria and bacteria in the morphogenesis of R. virens thalli”. The Cefalodia ontogeny (interactions between mycobiont hyphae and cyanobacteria) shows the process of organization of important structures “organs?” in the functionality of these lichen thalli. As well as the interactions of non-photosynthetical bacteriobionts being actively included and accumulated inside the lichen thallus, probably devising different functions in these micro-ecosystems (interactions between mycobiont hyphae and other bacteria). The same may be said of the structures of the pycnidia for asexual propagation of the fungus.

  1. line 19-20: “The present study was designed to contribute to a better understanding of this taxon, showing a clear decline over the last century”. I cannot find the data to show “a clear decline over the last century”. Please describe exactly what you found in this study and focus on the interpretation of the results.

This information cannot be cited in the Abstract, but it is clearly referenced in the Introduction (Lines 76-80), in the Discussion (Lines 483-485), and documented in the References list.

  1. Significant match-es of 100% - 99.85% identity and 95% - 236 100% coverage were obtained, re-spectively, with the S. reticulata strain SAG 53.87 237 (GU017650- GU053573). Aligned phycobionts LSU rRNA and nrITS DNA were 744 and 238 740 bp, re-spectively à Blast search is not enough to determine the identify of the photobionts. Please try phylogenetic analyses with reliable reference sequences and include the tree in supplementary materials like the mycobiont tree.

Thank you very much. Done. The new supplementary figure is Figure S1

In Material nad methods: “The second, the algal aligned ITS rDNA with selected sequences of Symbiochloris from the GenBank. Trebouxia jamesii (FJ626733) was included as the outgroup. The third, considered the cyanobacterial 16S….”

 “A multiple alignement… Akaike Information Criterion [60]: for R. virens ITS rDNA phycobiont GTR+G (Lines 178-180, 186-188)”,

In Results: “Aligned phycobionts LSU rRNA and nrITS DNA were 744 and 740 bp, respectively. All the sequences from nrITS DNA formed a well-supported clade (94/100) with the S. reticulata sequences included in this analysis. BI and ML phylogenetic hypotheses were topologically congruent. (Figure S1). (Lines 250-253),”

  1. Line 160-161: To perform cyanobacterial identification, the bacterial specific 16S rRNA gene 160 was amplified with the primer pair 740F [56] and 1494R [57]. à Lichen materials containdiverse bacterial species and it is expected that many species of bacterial rRNA genes areamplified. It is not clear how cyanobacterial sequences were selectively amplified or selected from clones or microbiome sequences. Please make it clear how cyanobacterial sequences were selectively obtained.

Thank you very much. Done. Lines 168-169

“To perform cyanobacterial identification, the bacterial specific 16S rRNA gene was amplified with the cyanobacteria specific primer pairs 740F [56] and 1494R [57]”

  1. Line 194-197: Several fragments of different thalli containing cephalodia were selected and fixed separately in: (a) a mixture of formaldehyde-acetic acid-ethyl alcohol (FAA) (to be used under light microscopy -LM-, scanning electron microscopy -SEM-, and low-temperature scanning microscopy – LTSEM-) and then stored in a refrigerator à Please make the sentence clearly understandable.

Thank you very much. Done. Lines 208-212

“Several fragments of different thalli containing cephalodia were selected to be observed under light microscopy -LM-, scanning electron microscopy -SEM- and low temperature scanning microscopy -LTSEM-. These samples were fixed in a formaldehyde-acetic acid-ethyl alcohol (FAA) mixture and subsequently stored refrigerated until use. Samples for TEM were fixed and dehydrated as described in Molins et al. (67).”

  1. Line 251, 253: Strain 1, Strain 2 à The word strain is usually used for an established pure culture. Please use more appropriate expression.

Thank you very much. Done. Changed to Nostoc sp. 1 and Nostoc sp.2 in the text and in the figures.

“two different Nostoc were detected. Nostoc sp. 1 was found in the two thalli from the population of VE and the thallus H10. According to GenBank data, Nostoc sp. 1 was previously observed in lichens such as Pannaria sp., Peltigera sp., and Ricasolia (Lobaria) (Figure S2). Nostoc sp. 2 was found”

  1. Citing references in the middle of the sentences disturb reading the paper. Please include citation at the end of the sentence unless it is very critical to cite in the middle of the sentence. For example,

Line 95: In Lobaria pulmonaria. [15,35] it was described that sometimes additional external

cephalodia may occur à In Lobaria pulmonaria, it was described that sometimes additional

external cephalodia may occur [15,35]

Line 434-436: In Lobaria pulmonaria, a coexistence consisting of a mycobiont [69,70], a

photosynthetic green alga 435 [Symbiochloris reticulata; 29,32] and a Nostoc cyanobiont [13,72]

is well known à In Lobaria pulmonaria, a coexistence consisting of a mycobiont, a

photosynthetic green alga (Symbiochloris reticulata) and a Nostoc cyanobiont is well known

(13,72, 29,32, 69,70).

Thank you very much. Done. See (Lines 99-100 and 473-475).

  1. Typolological errors should be corrected.

We have revised it along the text

  1. Discussion parts are majorly composed of data from other studies and only very little is to interpret the results of this study. à Please focus on interpretation of the results of this study.

The variety of techniques used in this work, and the amplitude and originality of some of the results achieved in a tripartite lichen that is vulnerable and that had not been studied in detail, required us to make an extensive literature survey in order to document and discuss them.

The Discussion was organized following the order outlined in the Results chapter: (

  1. a) background of the studies ;

(b) cyanobacterial constituents;

(c) mycobiont identification;

(d) phycobiont analyses, molecular and ultrastructural;

(e) thallus anatomy;

(f) ontogeny of cephalodia, herbarium observations;

(g) ultrastructural changes of cyanobacteria in symbiosis;

(h) ultrastructures related to the active incorporation of non-photosynthetic bacteria and interactions with hyphae of the upper cortex as well as considerations on the possible functions;

(i) conclusions.

Based on your remarks, we have tried to modify some paragraphs of the Discussion following your suggestions to emphasize the results of this study.

9.1. Paragraphs Lines 496-505-

9.2 Paragraph  Lines 512-516

9.3 Paragraph Lines 591-598

9.4 Paragraph Lines 681-682

9.5 Paragraph Lines 765-768

We needed to change the order of the references 91-96 to match the text